# Benefits of Table Tennis for Brain Health Maintenance and Prevention of Dementia

Takao Yamasaki [1,2,3]

1 Department of Neurology, Minkodo Minohara Hospital, Fukuoka 811-2402, Japan; yamasaki_dr@apost.plala.or.jp; Tel.: +81-92-947-0040
2 Kumagai Institute of Health Policy, Fukuoka 816-0812, Japan
3 School of Health Sciences at Fukuoka, International University of Health and Welfare, Fukuoka 831-8501, Japan

**Abstract:** Table tennis is an extremely popular sport throughout the world as it requires no expensive equipment, specialized amenities, or physical contact among players, and the pace of play can be adapted to allow participation by players of all skill levels, ages, and abilities. It is an aerobic-dominant sport driven primarily by the phosphagen system because rallies are relatively brief (several seconds) and separated by longer rest periods. Several studies have shown that physical interventions including table tennis can help prevent cognitive decline and dementia. Accordingly, the present paper provides an overview of the basic physical and cognitive demands of table tennis, reviews previous studies reporting improvements in physical and brain health across different non-clinical and clinical populations, and critically evaluates the usefulness of table tennis intervention for the prevention of cognitive decline and dementia. This review suggests that table tennis intervention could be a powerful strategy to prevent cognitive decline and dementia in the elderly.

**Keywords:** table tennis; aerobic energy system; phosphagen energy system; cognitive decline; dementia





## 1. Introduction

Table tennis (also known as ping-pong) is a racket sport played regularly by more 300 million people across all regions of the world, of whom at least 40 million are federated players [1–3]. The International Table Tennis Federation has the largest number of member countries (227) of any international sports federation [4], and table tennis has been part of the Olympic program since 1988 [5].

While the rules of table tennis are relatively simple and basic physical requirements minimal (i.e., there is no heavy equipment to manipulate and no physical contact), it requires a high level of concentration and hand-eye coordination to instantly predict and react to various rotations and trajectories of the ball. Table tennis is also a sport that can be enjoyed as entertainment because it can be played according to one's physical strength, age, skill, and purpose, and there are few injuries or accidents during play. Table tennis is thus both highly competitive and entertaining, and can be enjoyed by almost anyone [6].

Documented benefits of table tennis include improvements in hand-eye coordination, mental acuity, reflexes, balance, leg, arm, and core strength, and aerobic fitness; moreover, it provides a social outlet that may benefit mental as well as physical health [3,7]. Even recreational play has beneficial effects on body composition and lipid profiles in older adults [8]. In addition, table tennis participants report significantly higher life satisfaction and physical self-concept than non-exercisers [7]. In fact, it is reported that table tennis has a greater positive influence on cognitive function than other types of exercise [9], possibly due to the engagement of multiple muscle systems and brain networks. Several studies have also reported that regular play can be of great therapeutic benefit for individuals with chronic ischemic heart disease [10], Parkinson's disease [11], autism spectrum disorder [12], attention deficit hyperactivity disorder [13], and mild mental disabilities [14].

Currently, about 55 million people worldwide suffer from dementia, and this number is expected to reach 78 million by 2030 and 139 million by 2050 due to population aging in most industrialized countries and many developing nations. Dementia has deleterious effects on the physical, psychological, social, and economic status of the patient and also places a heavy burden on caregivers, families, and society [15]. Alzheimer's disease is the most common cause of dementia, accounting for an estimated 60–80% of all clinical cases [16]. Furthermore, mild cognitive impairment (MCI) is known as a pre-stage of dementia. In particular, amnestic MCI is widely considered a precursor to clinical Alzheimer's disease [17] and the total global population with MCI is larger and growing more rapidly than the Alzheimer's disease population. Therefore, there is an urgent need for interventions that prevent MCI and the progression of MCI to dementia.

Previous studies on physical activity interventions for patients with MCI and dementia have reported that improvements in physical health, especially aerobic health and fitness, are crucial for maintaining and enhancing brain health [18]. Notably, several such studies have reported that regular table tennis training can help maintain mental capacity and prevent or delay senile dementia [9,19]. Therefore, the present paper provides an overview of previous studies on the benefits of table tennis for physical and brain health, and critically examines the usefulness of table tennis for the prevention of cognitive decline and dementia.

## 2. Literature Search Strategy

In this paper, electronic searches for studies and information related to table tennis were conducted using PubMed, Google Scholar, CiNii, J-Stage, and Google Chrome (including all years). The following keyword combinations were used for the searches: "table tennis", "physical activity", "exercise", "brain", "cognitive function", "dementia", "dementia prevention", and "therapy". Inclusion and exclusion were decided based on the title and abstract of the paper as well as the contents of the website. Papers and websites not written in English were excluded, except for a pioneering paper written in Japanese (abstract written in English) [19] and the website of the Japan Table Tennis Therapy Association [20], which was established based on the same paper.

## 3. Characteristics of Table Tennis

### 3.1. The Concept of Table Tennis

Table tennis is a sport in which two or four players face each other across a small rectangular table (2.74 × 1.52 m) separated at the center by a net, and compete for points by hitting a plastic ball over the net to land on the other side (court) using a small racket [6] (Figure 1). Players repeatedly hit the light ball at high speeds at a close distance and so are required to track the movement of the ball, anticipate the trajectory, react appropriately, and hit the ball back toward the opponent's court within seconds to fractions of seconds. By controlling the spin, speed, and placement of the ball, players make it difficult for their opponents to successfully return the ball. Therefore, the player needs to predict their opponent's intentions, recognize meaningful cues in the context of the game, quickly decide on the best course of action, and generate an appropriate motor response [21]. Table tennis thus requires agility, hand-eye coordination, focused attention, foresight, and the ability to implement various techniques and tactics, thereby necessitating the simultaneous activation of multiple brain networks.

### 3.2. Game Dynamics

Table tennis competition is characterized by its intermittent but intense bouts of action (rallies) interspersed with brief periods of rest [2,22,23]. Participants receive one point for striking the ball back onto the opposition court without a successful return, and the winner is the first player to reach 11 points and win by at least 2 points. The total time required for a match ranges from 8 to 38 min, and some of the world's top players may play for as long as 45 min [1]. Matches are characterized by short rallies (i.e., physical and mental effort) lasting 3.4 s on average and longer pauses between rallies (i.e., rest periods)

lasting 11.6 s on average [24]. Each rally includes only 4.0 shots on average (or about 35.3 shots per minute). The length of the rally period is fairly consistent among players of different performance levels (3.2–3.6 s), whereas the pause period between rallies varies by performance level (i.e., 7.0 s for regional players, 9.3 s for national players, and 18.6 s for international athletes) [24]. As a result, the effort to rest ratio of regional players (0.5) is higher than that of national (0.34) and international athletes (0.18). These characteristics make table tennis a very intense sport, with the ball traveling at high speed (>50 km/h), forcing players to respond in milliseconds [2,23]. Consequently, agility, reaction time, ballistic strength, and coordination are essential skills that can be developed by regular table tennis practice [23].

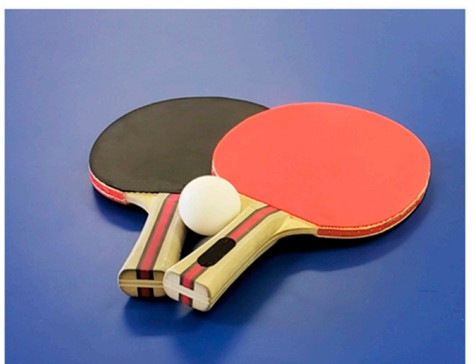 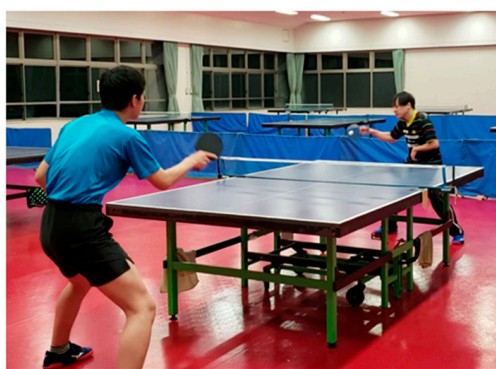

**Figure 1.** Table tennis rackets and ball (**left**), adapted from photo AC (https://www.photo-ac.com/, accessed on 13 July 2022), and two players engaged in a rally with the ball about to be struck in the far court (**right**).

### 3.3. Three Basic Energy Systems in Humans

To understand the physiological demands and characteristics of table tennis (Section 3.5.), it is first necessary to describe the multiple metabolic systems in humans. Energy for muscle activity in the form of adenosine triphosphate (ATP) is generated by three systems, phosphagen (ATP-creatine phosphate [CP]), anaerobic (glycolytic), and aerobic (oxidative) [25–27] (Figure 2). The phosphagen (ATP-CP) system uses CP and is characterized by a very high rate of ATP production. Due to the small amount of CP and ATP stored in muscle, however, the energy available for muscle contraction is limited. Nonetheless, it is sufficient for short-term, high intensity activities that last approximately 1–30 s. The anaerobic (glycolytic) system serves as a bridge between the acute phosphagen system and more sustained aerobic system. The anaerobic system does not require oxygen and uses the energy obtained by converting glucose to lactic acid in order to form ATP. This intermediate system can produce ATP quite rapidly for use during activities that require large energy bursts over a longer period of time (30 s up to 3 min) but at the cost of lactic acid accumulation. The aerobic (oxidative) system requires oxygen to produce ATP because carbohydrates and fats can be fully metabolized to $CO_2$ only in the presence of oxygen. These aerobic reactions occur in the cytoplasm and mitochondria of cells. The aerobic system produces ATP slowly, however, and is primarily used during prolonged lower-intensity activities after fatigue of the phosphagen and anaerobic systems. It is important that all three systems contribute to the energy needs of the body during physical activity, but one system will dominate at a given time depending on the duration and intensity of activity [25–27].

### 3.4. Measurement of Exercise Intensity

There are several ways to evaluate the exercise intensity (or amount of energy consumed) for a given sport, of which one of the simplest is by measuring parameters related to heart rate (HR), including mean HR (HRmean), maximum HR (HRmax), and %HRmax [28,29]. Another indicator of exercise intensity is maximal oxygen uptake ($VO_2$max), the maximum amount of oxygen that can be taken into the body per unit

time. This value determines the peak exercise intensity that an individual can tolerate over several minutes, so a higher $VO_2$max indicates greater endurance [29]. For comparison, exercise intensity is frequently expressed in metabolic equivalents (METs), where one MET represents the resting energy expenditure during quiet sitting and is commonly defined as a $VO_2$ of 3.5 mL/kg/min. The MET value for a given activity is the ratio of energy expended during that activity to energy expended at rest [30]. Blood lactate concentration is a suitable marker to establish the relative activities of the different energy systems during exercises and sports [1], and any sport or exercise can be ranked as light-, moderate-, vigorous-, or very hard effort according to %HRmax, %$VO_2$max, METs, and lactate concentration [31] (Table 1).

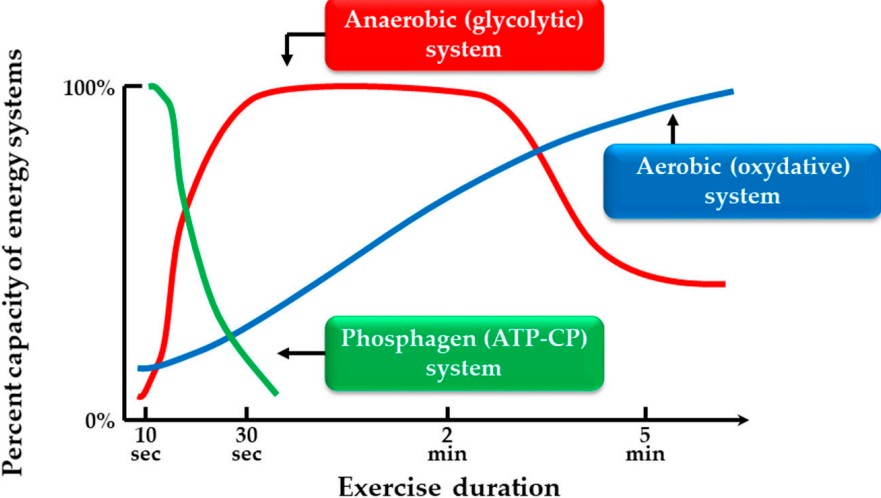

**Figure 2.** Three energy systems. Abbreviation: ATP-CP, adenosine triphosphate-creatine phosphate. The figure is adapted from Swanwick et al. [25] (CC BY NC 4.0).

**Table 1.** Measurement of exercise intensity. Adapted from Vanhees et al. [31].

| Intensity | %HRmax (%) | %$VO_2$max (%) | METs | Lactate (mmol/L) |
|---|---|---|---|---|
| Low intensity, light effort | 45–54 | 28–39 | 2–4 | 2–3 |
| Moderate intensity, moderate effort | 55–69 | 40–59 | 4–6 | 4–5 |
| High intensity, vigorous effect | 70–89 | 60–79 | 6–8 | 6–8 |
| Very hard effort | >89 | >80 | 8–10 | 8–10 |

Abbreviation: %HRmax, percent of maximum heart rate; %$VO_2$max, percent of maximal oxygen uptake; METs, metabolic equivalents.

### 3.5. Physiological Responses during Table Tennis

The physiological responses during table tennis matches were investigated in junior and adult players [22,32,33]. In adult table tennis players, reported HRmean was $142 \pm 11$ beats per minute (bpm), HRmax was $166 \pm 14$ bpm, $VO_2$max was $45.5 \pm 5.3$ mL/kg/min, and $VO_2$mean was $29.5 \pm 3.8$ mL/kg/min, corresponding to 66% of $VO_2$max. In addition, mean lactate was $1.4 \pm 0.4$ mmol/L and peak lactate was $1.8 \pm 0.6$ mmol/L [32]. In a study of young table tennis players, the HRmean was $164 \pm 14$ bpm, corresponding to $81.2\% \pm 7.4\%$ of the predicted HRmax, mean lactate was $1.8 \pm 0.8$ mmol/L, and peak lactate was $2.2 \pm 0.8$ mmol/L [22]. In junior table tennis players [33], the HRmean was $126 \pm 22$ bpm, HRmax was 189 bpm, $VO_2$mean was $25.6 \pm 10.1$ mL/kg/min, $VO_2$max was 45.9 mL/kg/min, mean METs was $4.8 \pm 1.4$, peak METs was 9.6, mean lactate was $1.1 \pm 0.2$ mmol/L, and peak lactate was 1.6 mmol/L [33]. The relatively low lactate accumulation during play suggests that the aerobic system is the principal energy source during pauses between rallies (rest periods) due to the long duration of matches, while the phosphagen system predominates during rallies (effort periods) [22]. Consistent with

these findings, a study analyzing the energetics of table tennis reported that 96.5% ± 1.7% of the overall energy expended is generated by the aerobic energy system with a minor contribution of 2.5% ± 1.4% from the phosphagen system and a negligible contribution (1.0% ± 0.7%) from the anaerobic system [34]. Another study found similar contributions of 96.6% ± 1.4% by the aerobic system, 2.3% ± 1.2% by the phosphagen system, and 1.1% ± 0.6% by the anaerobic energy system [32].

Taken together, energy demands in junior to adult table tennis players rely on the phosphagen system during the maximal short-duration effort of rallies (about 2% of the total energy expended) and the aerobic system during rest (pause) times (about 96% of the total energy expended) [32–35].

## 4. Benefits of Table Tennis for Physical and Brain Health

### 4.1. Benefits of Table Tennis for Physical Health

Several studies demonstrated the benefits of table tennis for body composition and physical fitness in children and adults [2,8,36–38]. For example, adult table tennis players presented higher fat-free mass and bone mineral density, and lower fat mass and body fat percentage [36], suggesting beneficial effects on a general health profile. Similarly, children who played table tennis regularly presented disparities in anthropometry and body composition compared to normally developed children who were not engaged in a regular physical activity or sport [2], including a lower body mass index, greater calf muscle perimeter, larger bone diameter, and greater bone mass. Children who played table tennis also showed superior fitness levels compared to normally developed but sedentary children, with greater aerobic capacity (i.e., VO$_2$max) and handgrip strength [2]. Among the elderly as well, recreational table tennis players exhibited higher total, regional (arm, leg and lumbar spine), and site-specific (trochanter and Ward's triangle) bone mineral density, lower total and regional (arm, leg and truck) fats mass, and lower percentage body fat compared to sedentary participants [37]. Furthermore, elderly table tennis players scored higher on a short physical performance battery and performed the 400-m walk in less time than age-matched sedentary participants [37]. These findings suggest that regular table tennis play has beneficial effects on muscle strength, physical performance, and body composition in older adults as well as younger adults and children. Regular play is also associated with improved performance indicators of daily life activities in the elderly [37]. Among older adults as well, serum high-density lipoprotein cholesterol ("good cholesterol") was higher and both lower low-density lipoprotein cholesterol ("bad cholesterol") and triglycerides were compared lower in regular recreational table tennis participants than age-matched sedentary participants [8]. Thus, recreational table tennis training can improve the serum lipid profile, an essential health index in older adults strongly associated with cardiovascular and neurovascular diseases.

A study investigating anthropometric profiles in table tennis players of various ages (i.e., senior, under-18, under-15, under-13, and under-11) found that both sexes exhibited <20% fat mass, while males showed ~45% lean mass and females ~37% lean mass [38]. Further, the healthy body composition status in children was maintained in older individuals who kept playing. Therefore, table tennis could be an effective activity for maintaining optimal health over the entire lifespan [38].

### 4.2. Benefits of Table Tennis for Brain Health

Regular physical activity not only improves cardiovascular health, but can also enhance cognitive function through neuroplastic changes [38–40]. Table tennis requires both large and fine motor control and sensory integration, leading to the activation and improved function of multiple neural regions and networks [39–41]. Functional near-infrared spectroscopy studies in adults have demonstrated extensive activation of motor-related areas such as primary motor cortex, premotor cortex, and inferior parietal cortex in experienced table tennis players compared to novices during play [39,40]. In addition, hemodynamic response magnitudes in these regions were positively correlated with the number of

strokes [39]. An electroencephalographic study of adults found greater spectral power of neural oscillations within the theta band (4–7.5 Hz) in frontal brain areas during table tennis compared to cycling and cognitive tasks [41], indicating that table tennis more effectively engages brain regions related to motor control, attentional processing, decision-making, and executive function.

Furthermore, long-term play can modify brain activity patterns even during other tasks, suggesting improved general neurological function [9,19,42–48]. Among adults over the age of 50, table tennis players obtained higher scores than non-players in the Kana Pick-out Test, a test of frontal lobe function used for dementia screening requiring subjects to simultaneously comprehend a written passage written in Kana characters while picking out selected vowels. In addition, there was a positive correlation between Kana Pick-out score and regularity of play [42], underscoring the benefits of table tennis for frontal lobe function. Similar results were obtained in another study using the Kana Pick-out Test on subjects from 10 to 70 years [19]. Moreover, another study found that table tennis improved mean score on the short-form Mini-Mental State Examination of general cognition among older adults compared to age-matched subjects that performed other physical activities [9]. Additionally, young table tennis players were found to score above average on all Delis–Kaplan Executive Function System tests, a battery measuring higher-level cognitive function (i.e., metacognition and executive function), compared to population norms [21]. In addition, a study examining the characteristics of attention network functions found improved executive control (but no difference in alerting or orienting network functions) in both young and adult table tennis players compared to age-matched non-athlete groups [43].

Electroencephalographic recordings while watching table tennis videos revealed stronger event-related desynchronization of the 8–10 Hz sensorimotor rhythm in the motor cortex of adult elite table tennis players compared to amateurs [44]. This finding suggests that greater motor skill increases the excitability of the motor cortex, possibly to facilitate reaction, movement planning, and execution under high attentional demands [44]. Another event-related potential study found that motor reaction time for visual motion was faster in young table tennis players than in age-matched non-athletes [45]. Specifically, latency of the N2 response (a negative component around 170 ms) originating from the visual motion sensitive area was significantly shorter in the table tennis players, indicating faster visual motion perception and processing speed [45].

A resting-state functional magnetic resonance imaging study of adults reported that the brain networks involved in attention control, visuomotor processing, and motor output were altered during table tennis skill progression from beginner to advanced [46]. Similarly, another functional magnetic resonance imaging study of adults performing a visuospatial task reported alterations in neural networks associated with the early processing of sensory information, next information integration, information matching identification, and late response selection induced by extensive table tennis training [47]. Collectively, these findings suggest that training can induce brain plasticity to enhance specialization and flexibility in the visuomotor systems of young or adult expert players [48].

In summary, table tennis can induce neuroplastic alterations in multiple brain networks including motor-related areas, visual cortex (in particular, visual motion area), and frontal regions, ultimately leading to improved sensorimotor and executive functions. Therefore, table tennis is an excellent physical activity for maintaining brain health. There is also accumulating evidence that these beneficial effects may prevent or delay cognitive decline and dementia in the elderly.

## 5. Effectiveness of Table Tennis Intervention in the Prevention of Cognitive Decline and Dementia

### 5.1. Physical Activity Interventions for the Prevention of Cognitive Decline and Dementia

World Health Organization guidelines recommend physical exercise, especially aerobic exercise, to prevent cognitive decline and dementia [49], a recommendation based

on numerous studies showing that physical activity interventions can prevent cognitive decline in healthy elderly people [18,50–54]. For example, a meta-analysis of aerobic exercise intervention studies concluded that improved fitness enhances cognitive function, especially in the domain of executive function [50], while a systematic review of randomized controlled trials spanning the adult lifespan revealed modest improvements in attention and processing speed, executive function, and memory following aerobic exercise interventions [51].

Aerobic exercise interventions may also improve memory among MCI patients [18,55,56]. A systematic review of randomized controlled trials specifically examining cognitively impaired individuals found improved global cognition, executive function, attention, and memory with increased physical activity [55]. This effect may be specific to the amnestic subtype of MCI as another study found that physical activity interventions failed to maintain cognitive function across all MCI subtypes, but significantly improved immediate memory from the baseline to the end of the 6-month interval in amnestic patients compared to controls [56].

In contrast, however, evidence for improved cognition by physical activity among dementia patients is mixed [18]. A meta-analysis found greater improvements in cognition relative to controls [57] and a recent umbrella review also concluded that physical activity/exercise has a positive effect on several cognitive and noncognitive outcomes in people with MCI and dementia [58]. However, other meta-analyses have found lesser or no benefits [59,60]. Therefore, as neurodegeneration progresses in dementia, it may be difficult to improve cognitive function through physical activity interventions alone, highlighting the importance of early physical activity intervention to delay cognitive decline in healthy elderly people, MCI patients, and possibly early-stage dementia patients.

*5.2. Mechanisms Underlying the Benefits of Physical Activity Interventions for Cognitive Decline and Dementia Prevention*

The benefits of physical activity on cognition are thought to be mediated by a variety of brain mechanisms, including improvement of cardiovascular risk factors, increased neurotrophic factor expression, amyloid-β turnover, and cerebral blood flow, and decreased inflammatory responses [18,61,62].

Cardiovascular risk factors such as diabetes, hypertension, hyperlipidemia, and obesity have been shown to cause hardening of cerebral blood vessels, small vessel damage, stroke, and reduced cerebral blood flow [18], cerebrovascular changes that may ultimately cause cognitive decline. Therefore, reducing cardiovascular risk factors through physical activity may be among the most effective strategies to prevent age-related cognitive decline.

Physical activity, especially aerobic exercise, also increases the expression of neurotrophic factors such as brain-derived neurotrophic factor (BDNF), insulin-like growth factor 1, and vascular endothelial growth factor [18,61–65]. BDNF is a neurotrophin involved in most important aspects of neuroplasticity, from neurogenesis to neuronal survival, from synaptogenesis to cognition, as well as in the regulation of energy homeostasis [63]. The increase in BDNF seems to correlate with the exercise volume [63] and is considered as a biomarker of exercise-induced cognitive benefits [64]. Open-skill exercise (e.g., table tennis, tennis, squash, basketball, or boxing) increases BDNF levels more than closed-skill exercise (e.g., running, swimming, cycling, golf, or archery), possibly because open-skill activities require additional attention to face over-changing situations, and are more enjoyable [63–65]. Insulin-like growth factor 1 and vascular endothelial growth factor play important roles in neurogenesis and angiogenesis and promote the expression of BDNF in the hippocampus [18]. Aerobic exercise training was found to increase the size of the anterior hippocampus in older adults, leading to improvements in spatial memory, and this increased hippocampal volume was associated with greater serum levels of BDNF [66]. Thus, increased expression of neurotrophic factors may be an important mechanism for the prevention of cognitive decline.

Amyloid-β plaques are a hallmark pathological feature of Alzheimer's disease [16]. In a longitudinal study of older adults, greater physical activity at baseline predicted lower plasma amyloid-β levels 9–13 years later. Furthermore, higher amyloid-β levels at year 9 predicted a greater risk of cognitive impairment at year 13, suggesting that amyloid-β levels also mediate the relationship between physical activity and cognitive impairment [67]. Similarly, a study using amyloid positron emission tomography found an inverse correlation between physical activity levels and brain amyloid-β load in older adults [68]. Thus, physical activity appears to promote amyloid-β turnover, which may contribute to the prevention of cognitive decline.

Cerebral blood flow decreases with age, which accelerates the decline in cognitive function and increases dementia risk in the general population [61]. Physical activity increases cerebral blood flow, which is thought to help maintain cerebral perfusion and prevent atrophy [62]. Several studies have also reported that regular physical activity can increase regional gray and white matter volumes, including in areas critical for memory, executive function, emotional regulation, and internally directed cognition such as the hippocampus, prefrontal cortex, and cingulate cortex [62].

Regular physical activity in elderly people has also been shown to reduce neuroinflammation as evidenced by lower serum concentrations of inflammatory markers such as C-reactive protein, interleukin-6, and tumor necrosis factor-α [61]. Furthermore, these decreases were associated with better performance in cognitive tests. Aerobic exercise lasting more than 2 weeks was found to improve immune system function in the healthy elderly by increasing the activity of natural killer cells as well as the proliferation of T lymphocytes, hematopoietic stem cells, and endothelial progenitor cells [61].

Taken together, physical activity and exercise (particularly aerobic exercise) contribute to the prevention and delay of cognitive decline and dementia through changes in the brain at the anatomical, cellular, and molecular levels [61].

*5.3. Table Tennis Intervention for Prevention of Cognitive Decline and Dementia*

Table tennis primarily involves moderate-intensity aerobic physical activity with a minor contribution by phosphagen energy generation (Section 3.5). Regular table tennis play can induce neuroplastic alterations in multiple brain networks, thereby sustaining or improving cognitive functions (Section 4.2) and potentially preventing age-related cognitive decline and dementia (Figure 3). A single-photon computed tomography study by Mori and Sato [19] also revealed that table tennis increased blood flow in motor-related areas, cerebellum, brain stem, and frontal lobe, and improved Revised Hasegawa Dementia Scale and Venton Visual Retention Test scores among brain disorder patients [19]. Based on this pioneering research, the Japan Table Tennis Therapy Association was established in 2014 [20]. Table tennis therapy for patients with dementia is also available in the United Kingdom [69] where the Bounce Alzheimer's Therapy Foundation founded in 2013 provides resources and training for table tennis-based therapy [69].

Despite multiple demonstrations of clinical efficacy [19,20,69], no studies have directly investigated the effects of table tennis interventions on brain functions linked to cognition, such as neurotrophic factor signaling and amyloid-β metabolism. To strengthen evidence for the prevention of cognitive decline and dementia, it is necessary to examine the effects of table tennis on the brain at the anatomical, cellular, and molecular levels. The promise of table tennis therapy for prevention of cognitive impairment and dementia is based on both the documented sensorimotor and cognitive engagement provided by this sport and the numerous randomized controlled trials demonstrating the benefits of combining exercise and cognitive training for cognitive improvement and reduction of brain atrophy in older adults with MCI [56,70–72].

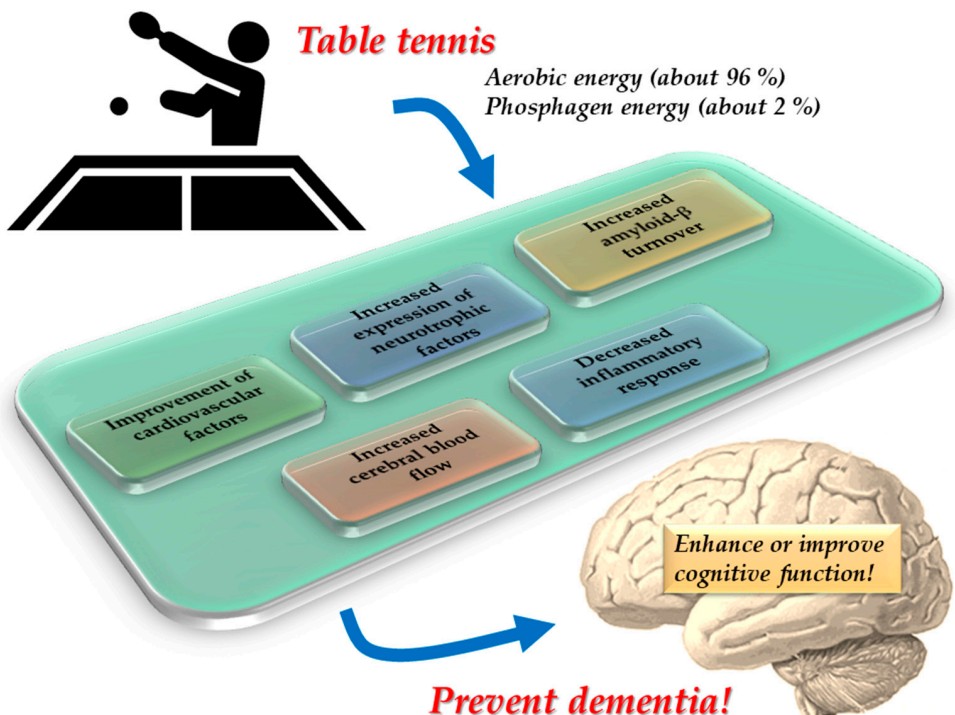

**Figure 3.** Possible mechanism of prevention of cognitive decline and dementia by table tennis.

A recent systematic review and meta-analysis demonstrated that low- to moderate-intensity exercise interventions without progression are most beneficial for female participants for improving cognitive function. In contrast, a progressive, very high-intensity exercise program can be expected to improve cognitive function in male participants [73]. To the best of my knowledge, no studies have examined gender differences in the effects of table tennis on cognitive function. However, given that table tennis is moderate-intensity physical activity, this intervention may be more effective in preventing cognitive decline and dementia in women than in men. Further studies are warranted to confirm this hypothesis.

Former president of the International Table Tennis Federation Mr. Ogimura has stated that "table tennis is like running the 100 m while playing chess at the same time" [74]. During table tennis practice or matches, players must not only perform aerobic exercise, but also judge the rotation, direction, and speed of the ball in milliseconds and instantly decide on how to react, cognitive tasks that require the simultaneous activation of multiple neural networks. In other words, table tennis is similar to an interventional method combining exercise and cognitive training and so may be more effective for preventing dementia than other aerobic exercises. In fact, one study found that table tennis exerted a greater influence on cognitive function than other types of exercise such as dancing, walking, gymnastics, and resistance training according to mean short-form Mini-Mental State Examination scores [9]. Thus, table tennis may be a more effective intervention for the prevention of cognitive decline and dementia than other aerobic exercises. However, further evidence is needed to confirm the efficacy of table tennis intervention programs.

## 6. Conclusions and Prospects

Table tennis is popular throughout the world because it can be played for recreation or competition by the majority of the population regardless of age, gender, or skill level. Regular play not only provides safe moderate-intensity aerobic exercise, but also induces neuroplastic changes in multiple brain networks underlying cognition. Therefore, table tennis is beneficial for both physical and brain health. For example, table tennis increases fat-free mass and bone mineral density, lowers fat mass and body fat percentage, and improves sensory processing, motor control, and executive functions. Furthermore, physical activity interventions using table tennis are clinically proven to prevent cognitive decline

and dementia in the elderly, and it has even been suggested that the preventive effect of table tennis is superior to other aerobic exercises. Accordingly, table tennis is among the best sports for maintaining physical and brain health into old age. However, several outstanding issues must be clarified to strengthen the evidence for cognitive decline and dementia prevention. First, no studies have directly investigated the effects of table tennis interventions on brain processes essential for the prevention of age-related cognitive impairments such as neurotrophic factor signaling and amyloid-β clearance, although clinical evidence is accumulating for such effects using various combined exercise and cognitive task interventions. Thus, it is necessary to examine the effects of table tennis on the brain at the anatomical, cellular, and molecular levels using neuroimaging and other modern methods. Second, it is also critical to verify that table tennis is superior to other aerobic interventions for cognitive decline and dementia prevention. For this purpose, a multicenter longitudinal study is desired. Finally, after addressing the first and second issues, it is necessary to create an optimal table tennis exercise program that is standardized across the globe.

**Funding:** This research received no external funding.

**Institutional Review Board Statement:** Not applicable.

**Informed Consent Statement:** Not applicable.

**Data Availability Statement:** Not applicable.

**Conflicts of Interest:** The author declares no conflict of interest.

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
