# Peer review of "Benefits of Table Tennis for Brain Health Maintenance and Prevention of Dementia"

_encyclopedia, doi:10.3390/encyclopedia2030107_

Round 1

Reviewer 1 Report

The present paper provides an overview of previous studies on the benefits of table tennis for physical and brain health, and critically examines the usefulness of table tennis for the prevention of cognitive decline and dementia.

The authors concluded that table tennis intervention could be a powerful strategy to prevent cognitive decline and dementia in the elderly.

The manuscript is well written and present interesting findings. However, some modifications are required:

All parts of the manuscript are well presented and present important findings and conclusion. However, I suggest that a methodological part for the paper is required to define:

-          The used keywords

-          How a study was included or excluded

-          The language of search

-          The databases

-          And all important information that could help readers

Author Response

To Reviewer 1:

Comment 1: All parts of the manuscript are well presented and present important findings and conclusion. However, I suggest that a methodological part for the paper is required to define: The used keywords, How a study was included or excluded, The language of search, The databases, And all important information that could help readers.

Response: Thank you for your valuable suggestion. According to your suggestion, I have added information on literature search strategy (P 2, Para 4).

Reviewer 2 Report

In this paper the author investigated the table tennis interventions to prevent cognitive decline and dementia in the elderly. The study was conducted with scientific rigor and following all the criteria that allow to obtain a good result about this topic. The topic is interesting and of value in the field and suitable to publish on Encyclopedia journal.

Nevertheless, the manuscript need minor revisions before its publication.

First of all, talking about the effects of this activity on brain health, I guess that is necessary cite the review performed from Di Liege CM et al 2019 from which it is also possible to take some sentences on the effect of physical activity on the production of BDNF.

The other thing, is the possibility to introduce a short discussion about the effect to prevent dementia talking about the difference in gender, if there are some evidence, and also if there is a correlation between the effect of this intervention and the previous physical activity that these elderly did in his life.

Author Response

To Reviewer 2:

Comment 1: First of all, talking about the effects of this activity on brain health, I guess that is necessary cite the review performed from Di Liege CM et al 2019 from which it is also possible to take some sentences on the effect of physical activity on the production of BDNF.

Response: Thank you for your valuable suggestion. According to your suggestion, I have cited paper you suggested and related papers, and described the relationship between physical activity and BDNF in more detail (P 8, Para 2, Lines 3–11).

Comment 2: The other thing, is the possibility to introduce a short discussion about the effect to prevent dementia talking about the difference in gender, if there are some evidence, and also if there is a correlation between the effect of this intervention and the previous physical activity that these elderly did in his life.

Response: Following your suggestion, I have added the discussion on gender differences in the effect of physical activity (including table tennis) on dementia prevention (P 9, Para 3–P 4, Para 1). 

Round 2

Reviewer 1 Report

After this revision, I suggest that the manuscript is suitable for publication in Encyclopedia.

Reviewer 2 Report

All comments has been implemented in the manuscript. No more comments. Thank you